# Functional Ensemble Distillation

**Coby Penso**
Bar-Ilan University, Israel
coby.penso24@gmail.com

**Idan Achituve**
Bar-Ilan University, Israel
idan.achituve@biu.ac.il

**Ethan Fetaya**
Bar-Ilan University, Israel
ethan.fetaya@biu.ac.il

## Abstract

Bayesian models have many desirable properties, most notable is their ability to generalize from limited data and to properly estimate the uncertainty in their predictions. However, these benefits come at a steep computational cost as Bayesian inference, in most cases, is computationally intractable. One popular approach to alleviate this problem is using a Monte-Carlo estimation with an ensemble of models sampled from the posterior. However, this approach still comes at a significant computational cost, as one needs to store and run multiple models at test time. In this work, we investigate how to best distill an ensemble's predictions using an efficient model. First, we argue that current approaches are limited as they are constrained to classification and the Dirichlet distribution. Second, in many limited data settings, all ensemble members achieve nearly zero training loss, namely, they produce near-identical predictions on the training set which results in sub-optimal distilled models. To address both problems, we propose a novel and general distillation approach, named Functional Ensemble Distillation (FED), and we investigate how to best distill an ensemble in this setting. We find that learning the distilled model via a simple augmentation scheme in the form of mixup augmentation [43] significantly boosts the performance. We evaluated our method on several tasks and showed that it achieves superior results in both accuracy and uncertainty estimation compared to current approaches.

## 1 Introduction

In parametric Bayesian inference the output is a distribution over model parameters, known as the posterior distribution. This is in contrast to standard approaches which return a single model. At test time one needs to marginalize over all models according to the posterior distribution to make a prediction. The action of averaging over multiple models allows Bayesian approaches to perform well with limited data [1, 2, 27, 34, 42] and properly estimate uncertainties [17, 18, 34]. The downside is that in most cases Bayesian inference is intractable and even methods for approximate inference may be costly. This is especially true for Bayesian neural networks (BNNs), as the dimension of the parameter space for common neural networks is very large [4, 15].

Two main approaches exist for efficient Bayesian inference. The first approximates the posterior distribution (e.g., variational inference and the Laplace approximation) with a simple distribution, usually a Gaussian with diagonal covariance. While this can be done efficiently in some cases [10, 20], it is more prone to over-fitting as the approximate distribution might cover only a small part of the posterior. A second approach is Monte-Carlo (MC) estimation where we sample $M$ models from the posterior using an approximate sampling algorithm, e.g. Markov Chain Monte-Carlo (MCMC), and estimate the mean prediction by the average over the predictions of the $M$ models. While this can better capture the diversity of the posterior, it requires storing and running $M$ models instead of one. One important question that will be the focus of this work is how to maintain the benefits of the Bayesian approach while reducing the computational overhead. It is also important to note that an ensemble of $M$ models is commonly used outside of Bayesian settings, e.g. bagging [3], so this line

36th Conference on Neural Information Processing Systems (NeurIPS 2022).

of work has wider implications. Our focus, however, will be in the low-data regime where Bayesian methods are most impactful.

We propose an ensemble distillation [16, 26] method that mimics an ensemble of models using a lightweight model. Current approaches mainly consider classification where a single prediction is a probability vector, the distilled model then outputs parameters of a Dirichlet distribution that should fit the ensemble predictions. Our approach, which we call Functional Ensemble Distillation (FED), on the other hand tries to learn a distribution over functions by transferring the randomness to $\epsilon$, a noise variable that is sampled from a fixed distribution. We designed our model with two requirements in mind. First, it should be fast to run during test time. This is achieved by being able to run with multiple $\epsilon$ values in parallel. Second, the distribution of predictions from our model should match the distribution of predictions from the posterior. This approach allows us more flexibility as it is not limited to classification tasks and to the Dirichlet distribution, unlike most previous approaches. Furthermore, our method allows us to compute covariance between predictions which can be useful for further downstream tasks. Consider, for example, the task of object detection for autonomous vehicles. If two nearby detections are made with certain confidence then the probability that that area of space is occupied depends heavily on the correlation between these predictions.

Finally, we investigate how to properly distill from a model ensemble under severe overfitting. It is common for deep networks trained on small datasets such as CIFAR-10 [22] to reach almost perfect accuracy on the training set while still generalizing well to the test set. In these cases, the predictions of the models in the ensemble can vary considerably on the test set despite being almost identical on the training set. This means that distilling the ensemble on the training set will not capture the diversity between predictions which is the core component of ensemble methods. While some previous works tried to address this issue [26, 29], they either depended on additional unsupervised data that might not be readily available, e.g. medical imaging, or do not scale well. Here we propose a simple solution for this problem that works remarkably well - creating an auxiliary dataset using the mixup augmentation [43]. The mixup procedure generates data that is on the one hand distinct enough from the training set to allow our ensemble to produce diverse predictions, while on the other hand, being close enough to it so that models that were trained on it work well when applied to the original test data.

To conclude, we make the following novel contributions: (i) We propose a general-purpose and flexible method for ensemble distillation that captures well the predictive abilities and the diversity of the original ensemble; (ii) we propose a simple, fast, and scalable method for obtaining a diverse auxiliary dataset for training the distilled model without *any* new samples using the mixup augmentation; and (iii) we evaluated our method on several image benchmarks and showed that it obtains the best results in terms of accuracy and calibration compared to other ensemble distillation approaches.

## 2    Background

We first provide a brief introduction to the main components of our model. We use the following notations in the paper. We denote scalars with lower-case letters (e.g., $x$), and vectors with bold lower-case letters (e.g., $\mathbf{x}$). We assume we are given with a dataset $\mathcal{D}$ of size $N$, $\mathcal{D} = \{(\mathbf{x}_i, y_i)\}_{i=1:N}$, where $\mathbf{x}_i \in \mathbb{R}^d$ are the training examples and $y_i$ are the corresponding labels or targets.

### 2.1    Bayesian Inference

In Bayesian inference we are mainly interested in the posterior distribution $p(\theta|\mathcal{D}) \propto \prod_i p(y_i|\mathbf{x}_i, \theta)p(\theta)$, where $\theta$ denotes the parameters of the model (e.g., a NN). The posterior tells us how likely $\theta$ is given the observed data. When given a new data point $\mathbf{x}_*$ we will use the entire posterior to calculate the predictive distribution using the rule $p(y_*|\mathbf{x}_*, \mathcal{D}) = \int p(y_*|\mathbf{x}_*, \theta)p(\theta|\mathcal{D})d\theta$. One intuition behind this is that when one has limited data, there are many distinct models that work well on the training set. By averaging over all the possible predictors the final prediction is supported by most of the "weight" in the posterior instead of by a single model. This also leads to better uncertainty estimation, as when many different models have different predictions the resulting average prediction is dispersed and uncertain.

In practice, besides a few notable exceptions, computing this integral is computationally intractable. One popular approximate inference solution is Monte-Carlo (MC) estimation. Namely, we estimate

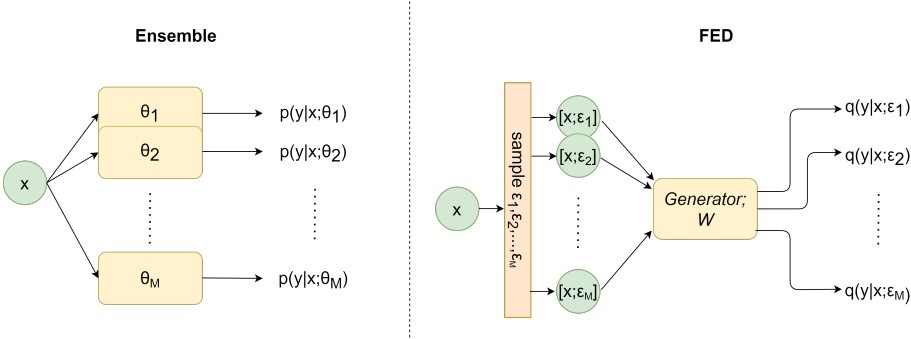

Figure 1: (1) Ensemble Forward pass pipeline. (2) FED Forward pass pipeline.

$p(y_*|\mathbf{x}_*, \mathcal{D}) \approx \frac{1}{M} \sum_{m=1}^{M} p(y_*|\mathbf{x}_*, \theta_m)$, where $\theta_m$ are (approximate) samples from the posterior $p(\theta|\mathcal{D})$. Obtaining such samples can be achieved in various ways such as MCMC methods [30, 38, 44]. One problem with this approach is that it requires storing and running multiple models to obtain the predictive distribution. This may be impractical in many situations such as when making predictions in low resource environments or in real-time systems.

## 2.2 Maximum Mean Discrepancy

A big question in machine learning and statistics is how to measure distances between distributions, specifically without access to the density function and with only access to samples. One well known measure is the Maximum Mean Discrepancy (MMD) [37] that was used as an objective to successfully train generative models [23, 24]. Given a family of functions $\mathcal{F}$ the MMD score between two distributions $p, q$ is defined as $MMD(p, q) = \sup_{f \in \mathcal{F}} |\mathbb{E}_{\mathbf{x} \sim p}[f(\mathbf{x})] - \mathbb{E}_{\mathbf{x} \sim q}[f(\mathbf{x})]|$. A specific case in which the MMD has a closed form formula is when $\mathcal{F}$ is the unit sphere in a reproducing kernel Hilbert space (RKHS) defined by a kernel function $k$ [13]. In that case we have

$$MMD(p, q)^2 = \mathbb{E}_{\mathbf{x}, \mathbf{x}' \sim p}[k(\mathbf{x}, \mathbf{x}')] - 2\mathbb{E}_{\mathbf{x} \sim p, \mathbf{y} \sim q}[k(\mathbf{x}, \mathbf{y})] + \mathbb{E}_{\mathbf{y}, \mathbf{y}' \sim q}[k(\mathbf{y}, \mathbf{y}')], \qquad (1)$$

which we estimate by Monte-Carlo means.

## 2.3 Mixup Training Procedure

The mixup [43] training procedure is based on the Vicinal Risk Minimization principle [6]. The main idea is to enlarge the training set by sampling points in the neighborhood of the given data. [43] proposed a simple way to achieve this goal by sampling along the line which connects two data points from the training set. Let $(\mathbf{x}_i, y_i), (\mathbf{x}_j, y_j)$ be two training examples, one can generate a new data point $(\tilde{\mathbf{x}}, \tilde{y})$ by taking their convex combination: $\tilde{\mathbf{x}} = \lambda \mathbf{x}_i + (1 - \lambda)\mathbf{x}_j$, and $\tilde{y} = \lambda y_i + (1 - \lambda)y_j$, where $\lambda$ is drawn from a Beta distribution with some fixed parameters. Now, standard loss functions, such as the cross entropy loss, can be used on $(\tilde{\mathbf{x}}, \tilde{y})$.

# 3 Related Work

Several previous studies proposed approaches to distill an ensemble of models into a single fast model. Early approaches [5, 16, 21, 32, 40] focused on capturing only the mean behaviour of the ensemble but did not capture the diversity in it, which may be crucial for uncertainty estimation. For example, in [5] it was first shown that a single network can learn the knowledge of a large ensemble of networks. In [16] a distilled model is learned using a convex combination of the ensemble predictions in addition to the standard cross-entropy loss. And, in [21] a student network is learned online from a teacher that is trained via MCMC training procedures.

More recent studies proposed alternatives for capturing the diversity as well. Ensemble Distribution Distillation (EnDD) [26] was the first to propose such an approach. EnDD builds on the method in [25] which tries to emulate the behaviour of an ensemble with a Dirichlet distribution parameterized by a NN. To handle the fact that the ensemble predictions are over-confident and have low diversity

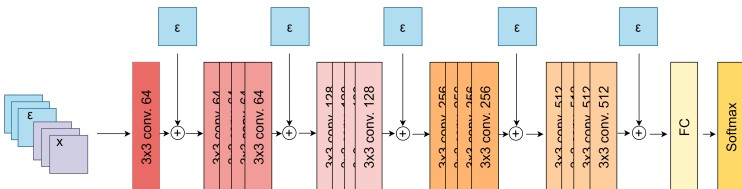

Figure 2: Generator architecture. First, we concatenate 3 random Gaussian channels to the input. Second, add Gaussian noise to intermediate features along the ResNet-18 architecture.

on the training set, they proposed to use an unsupervised auxiliary dataset for training the distilled model, and to perform temperature annealing. Similarly, [9] also makes the assumption that the predictive distribution follows a Dirichlet distribution. They fit the output to the teacher network by two separate networks, one that learns the predicted class probabilities, i.e. the Dirichlet mean, and another that learns the confidence, i.e. the Dirichlet concentration. Both [26, 9] are suitable for classification tasks only and assume a Dirichlet distribution behaviour which may be too restrictive. We argue that a more general solution for the distillation of ensembles is needed.

A different line of work [36, 29] generalizes the method in [16] by distilling the ensemble members to task heads in a one-to-one fashion. In [36] the distilled model is partitioned into two parts, a shared "body" and a member-specific "head", one per ensemble member. Each head is trained to copy the prediction of an ensemble member using the standard cross-entropy loss function. In [29] the authors propose to use BatchEnsemble [39] for weight sharing. They increase the diversity in the ensemble predictions on the training set by randomly perturbing the training data based on three techniques: Gaussian, Output diversified sampling (ODS) [35], and ConfODS. The ODS perturbations are similar to adversarial perturbations and need to be computed at every iteration. This imposes a computational overhead which renders a slow training process and inability to easily handle large ensembles, unlike our proposed method.

## 4 Our Method

We take a similar approach to [9, 26] where we treat the ensemble distillation as a conditional generative model. Given an input $\mathbf{x}$ and a sampled model parameters $\theta$ we get a distribution $p(y|\mathbf{x}, \theta)$, so for a given $\mathbf{x}$ we can push-forward the posterior distribution over weights to a distribution over distributions over $y$. While this was done in [9, 26], there are several limitations with these approaches. First, they use a Dirichlet distribution to model this distribution over distributions, which is limited to classification. Second, even in classification settings, the family of Dirichlet distributions has fixed expressive power, and there is no evidence that the posterior is indeed a Dirichlet distribution.

Our approach is to model a distribution over functions that both have low memory requirements and that we can parallelize to get multiple predictions quickly. We do this by training a generator neural network with parameters $\mathbf{w}$ and added random noise $\epsilon$. The noise can be added, for example, as an additional channels to the input or hidden layers. Our goal is to train the generator network so that the distribution over $q(y|\mathbf{x}, \mathbf{w}, \epsilon)$ for $\epsilon \sim p(\epsilon)$ will match that of $p(y|\mathbf{x}, \theta)$ for $\theta \sim p(\theta|\mathcal{D})$. Namely, each draw of $\epsilon$ corresponds to a draw from the posterior. By using this design we get a similar memory cost to a standard single network and we can parallelize easily as all the linear operations are shared. Computationally it is equivalent to running a single model on a batch instead of a single input. An illustration of our method is presented in Figure 1. We further show in the appendix that for a wide range of batch sizes the runtime of our model scales approximately as $1 + (\# \text{ of } \epsilon \text{ samples})/35$ resulting in a considerable speedup. Since we now have a distribution over functions, we call our method *Functional Ensemble Distillation (FED )*.

We now present our training objective for the generator $g_{\mathbf{w}}$. Given an ensemble member $\theta_j$ and a batch of $B$ examples, we consider the vectors of predicted probabilities (one per example) as features representing the function parameterized by $\theta_j$. Denote by $\mathbf{p}_i^j = p(y_i|\mathbf{x}_i, \theta_j)$ the predictive likelihood for the $i^{th}$ example by the $j^{th}$ function. We represent the function parameterized by $\theta_j$ by concatenating these vectors to form a feature vector $\hat{\mathbf{p}}^j = concat(\mathbf{p}_1^j, ..., \mathbf{p}_B^j)$. We repeat this process for our generator network (only that now each $\epsilon_i$ represents a function) and in a similar fashion, we

---

**Algorithm 1** Generator training

---

1: **input:** Predictions of the ensemble on our distillation dataset: $\{\mathbf{x}_i, \{\mathbf{p}_i^j\}_{j=1:M}\}_{i=1:N}$
       # N number of data points, M ensemble predictions
2: Initialize Generator $g_{\mathbf{w}}$
3: **while** not converged **do**
4:     Sample batch of inputs and model predictions $\{\mathbf{x}_i, \{\mathbf{p}_i^j\}_{j=1:M}\}_{i=1:B}$
5:     Sample $M$ latent variables - $\epsilon \sim \mathcal{N}(0, \sigma^2)$
6:     $\mathbf{q}_i^j = g_{\mathbf{w}}(\mathbf{x}_i, \epsilon_j)$
7:     $\hat{\mathbf{q}}^j = concat(\mathbf{q}_1^j, ..., \mathbf{q}_B^j)$
8:     $\hat{\mathbf{p}}^j = concat(\mathbf{p}_1^j, ..., \mathbf{p}_B^j)$
9:     $\mathcal{L} = \widehat{MMD}^2(\{\hat{\mathbf{p}}^j\}_{j=1:M}, \{\hat{\mathbf{q}}^j\}_{j=1:M})$
10:    $\mathbf{w} \leftarrow \mathbf{w} - \nabla_{\mathbf{w}}\mathcal{L}$
11: **end while**
12: **return** $g_{\mathbf{w}}$

---

obtain a representation $\hat{\mathbf{q}}^j = concat(\mathbf{q}_1^j, ..., \mathbf{q}_B^j)$ using the same batch. The same process is applied to every member of the ensemble. Now we use the MMD objective presented in Section 2.2 to score the difference between distribution over functions, using a simple kernel over these functions' representations. More formally, the MMD objective given a batch of examples is:

$$\widehat{MMD}^2\left(\{\hat{\mathbf{p}}^j\}_{j=1:M}, \{\hat{\mathbf{q}}^j\}_{j=1:M}\right) = \frac{1}{M^2}\sum_{i,j}\left(k(\hat{\mathbf{p}}^i, \hat{\mathbf{p}}^j) + k(\hat{\mathbf{q}}^i, \hat{\mathbf{q}}^j) - 2k(\hat{\mathbf{p}}^i, \hat{\mathbf{q}}^j)\right). \quad (2)$$

We present our method in Algorithm 1. In our experiments, we used standard convolutional neural networks where we concatenated 3 random Gaussian channels to the input channels and we add Gaussian noise to intermediate features across the network. We evaluated several options for inserting noise into the generative model and noticed that to learn sufficient diversity, enough noise has to be inserted; otherwise, the generator's diversity suffers and collapses to a nearly deterministic model. We also found it best to set the noise STD to be a learnable parameter (one parameter per input noise layer). See an illustration with ResNet18 in Figure 2. Regarding the limitations of our approach, we note that while the run-time is much faster then running the full ensemble, there is still a non-negligible overhead compared to a single model.

## 4.1 Auxiliary Dataset

One problem with distilling ensemble models is that the predictions of the ensemble members on the training set have a much higher agreement compared to the test set (as can be seen in Table 1). Practically this means that distillation of the ensemble predictions on the training set can lead to a degenerate solution that returns in most cases the same predictions. This leads to worse performance in both accuracy and calibration compared to the original ensemble.

In [26] the authors address this by using an auxiliary unsupervised dataset, as we can get diverse ensemble predictions on this dataset and use it to train our distilled model. While this solution performs quite well, in several applications having access to an additional unsupervised dataset is not trivial. One example is medical imaging [8], where even unsupervised data requires access to expensive machinery and data that is not readily shared due to privacy concerns. Another example is learning from ancient texts [12] where digitized data, labeled or unlabeled, is scarce. We argue that these problems in which the data is limited are exactly the problems where Bayesian methods can make the most impact. Therefore, having a method that works on these types of data is of great importance.

This leads us to seek a solution that does not require any additional data, even if it is unsupervised. We investigated several solutions (see Section 5.4) and found one simple solution that worked surprisingly well, the mixup augmentation. The predictions on the mixup data are more diverse allowing us to distill from data that more closely resembles the test data. Another important property of the mixup augmentation is that models trained on it transfer well to real images. This is in contrast to other

alternatives that we explored, such as generating an auxiliary dataset using a generative adversarial network (GAN), that did not transfer well.

To summarize we generate a synthetic, unlabeled, dataset from the original training set (that was used to train the original ensemble) using the mixup augmentation. We then train our generator network using the predictions of the original ensemble on this dataset, see Algorithm 1. In Table 3, we show that indeed the ensemble classifier generates a more diverse set of predictions on the mixup dataset compared to the actual training set.

# 5    Experiments

We evaluated FED against baseline methods on different datasets and using different metrics. We present here the results of the models using the hyperparameters with the best accuracy on the validation set. As a result, for some models the diversity can be improved at the expense of lower accuracy. We will also provide our code for reproducibility - `https://github.com/cobypenso/functional_ensemble_distillation`.

**Datasets.** We evaluated all methods on CIFAR-10, CIFAR-100 [22], and STL-10 [7] datasets. STL-10 and CIFAR-100 are considered more challenging, the former has a relatively small amount of labeled training data (10 classes with 500 labeled examples per class), and the latter has more categories. For all datasets we use train/val/test split. The train/val split with a ratio 80%:20%.

**Compared methods.** We compare our method against the following baselines: (1) *EnDD* [26], Ensemble Distribution Distillation framework, trained with the same train data used to capture the ensemble. (2) *EnDD$_{AUX}$* [26], Ensemble Distribution Distillation framework, trained with the train data and an auxiliary dataset. For CIFAR-10, CIFAR-100 is the auxiliary dataset. For CIFAR-100, CIFAR-10 is the auxiliary dataset. For STL-10 we used the unlabeled data provided as part of this dataset. We stress that other methods do not have access to this data. (3) *DM* [29], this baseline comes in three variations, *Gaussian*, *ODS*, and *ConfODS*. (4) *AMT* [9], a method which also uses MMD as an objective. (5) *Hydra* [36], a weight sharing method based on [16]. (6) Knowledge Distillation (*KD*), a naive single model baseline, trained on the mean predictions on the mixup data.

**Ensemble.** We created the Bayesian ensemble (for distillation) by sampling from a posterior obtained with the Cyclic Stochastic Gradient Hamiltonian Monte Carlo (cSGHMC) method [44]. Overall we sampled 120 models. In addition, we show experimented with different ensemble methods and their implications on the distillation. The architecture we choose for the models in the ensemble is ResNet18 with Group Norm [41] and Weight Standardization [33]. We did not use Batch normalization since it breaks the IID assumption commonly used in Bayesian Inference [17].

We note that sampling from the posterior with cSGHMC, instead of SGD, slightly improves the ensemble diversity on the training set. If our ensemble was sampled using SGD with different seeds the effect of mixup would be even more pronounced.

**Training setup.** For the baselines, we follow the training strategies suggested in the papers and their code if available, while performing an extra hyperparameter search when training on datasets which did not cover in the original papers, mainly CIFAR-100 and STL-10. For our method, we performed a standard training procedure with Adam optimizer [19], a learning rate scheduler with fixed milestones at epochs $\{35, 45, 55, 70, 80\}$, and a hyperparameter search done over a held-out validation set.

**Objective and Kernel choice.** We trained our model using the MMD objective while considering various kernels. Specifically, we examined Linear kernel, RBF kernels, and a mixture of RBF kernels. For the RBF kernel, we considered different values of length scales, as we saw that it had a considerable effect on the results. For CIFAR-10, a mixture of RBF kernels with $\{2, 10, 20, 50\}$ length scales had the best results. For CIFAR-100, the length scales are $\{10, 15, 20, 50\}$, and for STL-10 a length scale of 50 works best.

**Distilled Model.** We distilled using a similar ResNet18 with Group Norm and Weight Standardization architecture as the ensemble models. Thus, essentially trying to distill a 120 models ensemble into a single model that has almost the same size as a member in the ensemble, i.e a $\sim 100\times$ memory improvement. Memory and Timing performance analysis can be found in the Appendix. For the noise sampled and fed into the generative model, we chose a simple Gaussian distribution, fed by

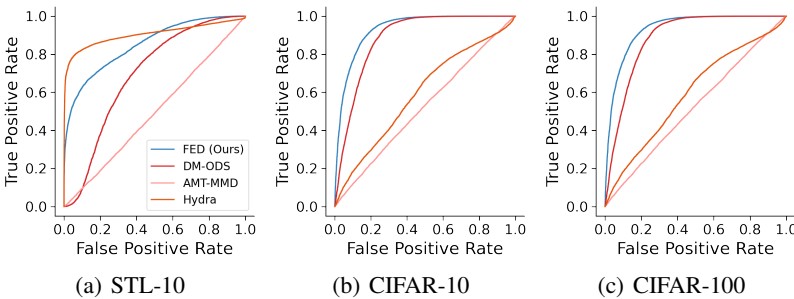

|  | (a) STL-10 | (b) CIFAR-10 | (c) CIFAR-100 |

Figure 3: Receiver operating characteristic (ROC) curve measuring the performance on in-distribution vs out-of-distribution data based on the mutual information. Best viewed in color.

concatenating it to the input, and by inserting it in intermediate places along the model. Specifically, for the concatenation part, 3 channels of noise and 3 channels of the input are stacked together. For the intermediate noise, the Gaussian noise was added to the features, instead of concatenation, in 5 different places, one after the first convolution layer and the other four after each Block in the ResNet-18 architecture (Figure 2). We found that setting the standard deviation of the noise to be a learnable parameter as part of the optimization, has a slight performance improvement.

**Metrics.** To evaluate our method, and other baselines, we define the agreement metric as the probability that two distinct elements of our ensemble predict the same label on the same input. In addition, we computed accuracy, and Expected Calibration Error (ECE) [14] reported in percentages. For the agreement metric, a close value to the original ensemble's agreement is a good indication that we capture similar behavior.

## 5.1 Ensemble Analysis

Table 1: Ensemble analysis. Test accuracy, agreement & ECE (%) on STL-10, and CIFAR-10/100.

|  | STL-10 | | | CIFAR-10 | | | CIFAR-100 | | |
|  | ACC (↑) | AGREEMENT | ECE (↓) | ACC (↑) | AGREEMENT | ECE (↓) | ACC (↑) | AGREEMENT | ECE (↓) |
|---|---|---|---|---|---|---|---|---|---|
| Train | 100.0 | 98.23 | 2.99 | 99.97 | 91.85 | 6.38 | 99.95 | 97.53 | 4.15 |
| Test | 81.84 | 87.85 | 2.17 | 94.41 | 86.08 | 5.56 | 77.79 | 71.97 | 3.07 |
| mixup | – | 95.20 | – | – | 88.08 | – | – | 88.56 | – |

We first analyze the ensemble we sampled using cSGHMC. To evaluate the diversity between predictions we use the agreement metric. We computed accuracy, agreement and Expected Calibration Error (ECE) [14, 28] over the trainset and testset. We only show agreement on our suggested mixup auxiliary dataset, as it does not have discrete labels. We can see that the ensemble has almost perfect accuracy and very high agreement on the training set, with a considerable drop in both on the testset. In fact, for CIFAR-100 and STL-10, the ensemble has essentially no diversity over the training set, thus making the trainset inappropriate for the distillation task. The mixup agreement closes part of the gap between the train and test datasets.

## 5.2 Distillation

We now evaluate FED on learning multi-class classification tasks. While we are not limited to classification tasks, In the following experiment we compare our approach to the baselines described above. Results in Table 2 show that FED outperforms previous methods that do not use an auxiliary dataset by a large margin, especially on the more challenging STL-10 and CIFAR-100 datasets. This is true for both accuracy and ECE. It is interesting to note that while EnDD$_{AUX}$ uses additional unlabeled examples from the training distribution (between $50k$ and $100k$), FED is on par and often outperforms this baseline across all metrics, and is the only one who outperforms the naive single model distillation baseline (KD) on two out of three datasets.

We also show results for out-of-distribution (OOD) detection in Figure 3 using ROC curves (EnDD was excluded due to poor performance), To score the examples we used the knowledge uncertainty

Table 2: Test accuracy, agreement & ECE (%) (± STD) on STL-10 and CIFAR-10/100. Note that the EnDD$_{AUX}$ baseline uses additional unsupervised data. Each experiment was repeated 3 times.

| | STL-10 | | | CIFAR-10 | | | CIFAR-100 | | |
|---|---|---|---|---|---|---|---|---|---|
| | ACC (↑) | AGREEMENT | ECE (↓) | ACC (↑) | AGREEMENT | ECE (↓) | ACC (↑) | AGREEMENT | ECE (↓) |
| Ensemble | 81.84 | 87.85 | 2.17 | 94.41 | 86.08 | 5.56 | 77.79 | 71.97 | 3.07 |
| KD | 80.49±.12 | – | 4.15±1.1 | 91.10±.21 | – | 3.53±.02 | 73.45±.00 | – | 3.83±.64 |
| EnDD$_{AUX}$ | **80.78**±**.00** | 99.98±.00 | **1.20**±**.00** | 91.05±1.3 | 100.0±.00 | 5.63±1.0 | 70.68±.00 | 100.0±.00 | 21.40±.00 |
| EnDD | 73.45±1.1 | 99.88±.02 | 10.62±1.2 | 91.01±.70 | 99.99±.00 | 6.39±.81 | 71.16±.00 | 100.0±.00 | 14.51±.00 |
| DM$_{Gaussian}$ | 74.21±.13 | 86.63±3.4 | 15.17±.58 | 90.27±1.5 | 95.88±1.7 | 2.48±.94 | 72.44±.19 | 85.97±.81 | 4.63±1.0 |
| DM$_{ODS}$ | 73.90±.65 | 86.72±4.3 | 15.32±.88 | 89.35±2.7 | 95.51±1.7 | 2.65±1.2 | 71.85±.36 | 87.90±4.1 | 4.47±1.0 |
| DM$_{ConfODS}$ | 73.26±2.2 | 86.67±2.7 | 14.52±3.1 | 89.55±2.3 | 95.49±2.3 | 2.59±.29 | 72.33±.54 | 88.73±4.7 | 5.33±1.1 |
| AMT$_{MMD}$ | 54.36±.78 | 11.31±.08 | 38.97± .68 | 83.81±.56 | 12.95±.08 | 64.00± .63 | N/A±N/A | N/A±N/A | N/A±N/A |
| Hydra | 75.74±1.6 | 93.40±.26 | 30.68±3.3 | 91.07±1.8 | 94.60±.62 | 5.39±.61 | 69.56±.11 | 93.80±.46 | 29.84±1.0 |
| FED (Ours) | **80.61** ± **.36** | 97.18 ± .06 | 3.41 ± .94 | **93.61** ± **.06** | 93.34 ± .34 | **0.66** ± **.10** | **74.48** ± **.10** | 87.82± .60 | **1.64** ± **.17** |

which is obtained by subtracting the aleatoric uncertainty from the total uncertainty [11, 26]. We used SVHN [31] as OOD data. See appendix for more details. From the figure, FED generally outperforms all baselines here as well.

## 5.3 Mixup Augmentation as Auxiliary Dataset

Table 3: Ablation - EnDD, KD and Fed with and without mixup. Test accuracy, agreement & ECE (%) on STL-10, CIFAR-10, and CIFAR-100.

| | STL-10 | | | CIFAR-10 | | | CIFAR-100 | | |
|---|---|---|---|---|---|---|---|---|---|
| | ACC (↑) | AGREEMENT | ECE (↓) | ACC (↑) | AGREEMENT | ECE (↓) | ACC (↑) | AGREEMENT | ECE (↓) |
| EnDD | 72.23 | 99.89 | 11.96 | 90.35 | 100.0 | 6.80 | 71.16 | 100.0 | 14.51 |
| EnDD + mixup | 78.40 | 99.99 | 7.68 | 90.83 | 100.0 | 5.74 | 68.78 | 100.0 | 15.86 |
| KD | 64.60 | – | 16.13 | 90.78 | – | 9.90 | 73.45 | – | 3.83 |
| KD + mixup | 80.49 | – | 4.15 | 91.10 | – | 3.53 | 73.96 | – | 13.72 |
| FED | 65.06 | 97.18 | 5.00 | 92.58 | 99.16 | 1.92 | 68.34 | 97.58 | 9.61 |
| FED + mixup | **80.94** | 97.16 | **2.55** | **93.68** | 93.71 | **0.78** | **74.56** | 88.20 | **2.07** |

As an ablation study, we would like to understand the separate effects of the mixup augmentation and our FED approach. Thus, we show results for both FED, KD and EnDD trained on the training set and on the mixup dataset as the auxiliary dataset. Results are detailed in Table 3. We show that for our method the mixup adds a significant boost to performance, and also for EnDD and KD on STL10. We also see that even when EnDD is trained with mixup, our approach returns better accuracy, ECE and also has greater diversity. We suspect that this is due to the limited expressive power of the Dirichlet distribution. Additional experiments in the Appendix show that at least part of the lack of diversity can be attributed to the Dirichlet distribution.

## 5.4 Alternatives for Learning Distilled Model Without Auxiliary Dataset

In this section, we explore an additional attempt to distill the ensemble, this time without access to additional data. This approach was inspired by the Out-Of-Bag (OOB) estimate in bagging. In bagging each model is trained only on a random subset of the training set and the OOB estimate is done by averaging the predictions of all models on those data points that they were not trained on and returning their score (which we refer to as held-out datasets). Using this approach, we can train each member in the ensemble on part of the dataset, and use for distillation only the ensemble members' predictions on the data points that they were not trained on. As an example, consider a single datapoint $\mathbf{x}$ which is on the held-out dataset for model $m_1$, but is part of the trainset for model $m_2$. To train the distillation model, the ensemble uses only the predictions of model $m_1$, but does not use those of model $m_2$ to get the overall output for $\mathbf{x}$. We experimented with two different partitions for held-out datasets: K-fold and bagging.

In K-Fold we first split the data into $K = 10$ folds, $\mathcal{D}_1, ..., \mathcal{D}_{10}$. For each of the 10 folds we define $\mathcal{D}_{-j} = concat(\{\mathcal{D}_i\}_{i \in \{1,...,10\}, i \neq j})$. We then sample 12 models using cSGHMC on each $\mathcal{D}_{-j}$ dataset. As before we get 120 models and for each fold $\mathcal{D}_j$ we can use the held-out models, i.e. models trained on $\mathcal{D}_{-j}$, for distillation. Note that the dataset for distillation is the entire trainset, but for each data-point, we use only the predictions of 12 models that did not train on it.

Table 4: Comparison of ensemble techniques on STL-10, CIFAR-10, and CIFAR-100.

| | | STL-10 | | | CIFAR-10 | | | CIFAR-100 | | |
| --- | --- | --- | --- | --- | --- | --- | --- | --- | --- | --- |
| | | ACC ($\uparrow$) | AGREEMENT | ECE ($\downarrow$) | ACC ($\uparrow$) | AGREEMENT | ECE ($\downarrow$) | ACC ($\uparrow$) | AGREEMENT | ECE ($\downarrow$) |
| **Ensemble** | Train | 100.0 | 98.22 | 2.99 | 99.96 | 91.85 | 6.38 | 99.94 | 97.53 | 4.15 |
| | Test | 81.85 | 87.85 | 2.17 | 94.41 | 86.08 | 5.56 | 77.79 | 71.97 | 3.07 |
| | mixup | – | 95.20 | – | – | 88.08 | – | – | 88.56 | – |
| **10 Fold** | Train | 96.95 | 89.47 | 12.45 | 99.99 | 97.78 | 1.77 | 99.97 | 94.08 | 4.77 |
| | Held-out | 76.67 | 91.86 | 1.42 | 93.72 | 93.42 | 1.21 | 74.52 | 78.95 | 3.67 |
| | Test | 80.48 | 80.45 | 5.85 | 94.84 | 91.75 | 1.48 | 77.70 | 72.81 | 2.43 |
| | mixup | – | 85.49 | – | – | 94.08 | – | – | 86.61 | – |
| **10 Bagging** | Train | 97.32 | 82.05 | 14.71 | 99.70 | 94.24 | 3.74 | 99.48 | 79.79 | 14.42 |
| | Held-out | 76.90 | 79.03 | 2.39 | 91.43 | 91.43 | 0.94 | 73.27 | 69.32 | 3.27 |
| | Test | 78.85 | 74.04 | 5.63 | 93.99 | 90.03 | 1.70 | 74.78 | 65.53 | 5.21 |
| | mixup | – | 78.01 | – | – | 90.83 | – | – | 73.59 | – |
| **120 Bagging** | Train | 96.33 | 77.84 | 18.85 | 99.99 | 94.63 | 3.59 | 99.98 | 79.97 | 14.08 |
| | Held-out | 77.79 | 71.85 | 8.59 | 93.85 | 91.69 | 0.81 | 74.68 | 69.13 | 3.77 |
| | Test | 77.77 | 70.49 | 8.57 | 94.13 | 91.09 | 1.12 | 75.16 | 67.68 | 4.02 |
| | mixup | – | 73.86 | – | – | 91.52 | – | – | 74.19 | – |

Table 5: Comparing FED + mixup and Out-Of-Bag ensembles on a held-out data. Test accuracy, agreement and ECE (%) on STL-10, CIFAR-10, and CIFAR-100.

| | STL-10 | | | CIFAR-10 | | | CIFAR-100 | | |
| --- | --- | --- | --- | --- | --- | --- | --- | --- | --- |
| | ACC ($\uparrow$) | AGREEMENT | ECE ($\downarrow$) | ACC ($\uparrow$) | AGREEMENT | ECE ($\downarrow$) | ACC ($\uparrow$) | AGREEMENT | ECE ($\downarrow$) |
| Bayesian$_{mixup}$ | **80.94** | 97.16 | **2.55** | **93.68** | 93.71 | **0.78** | **74.56** | 88.20 | 2.07 |
| 10-Fold$_{Held-out}$ | 62.92 | 84.40 | 5.33 | 91.00 | 90.48 | 0.87 | 67.73 | 80.39 | **1.62** |
| 10-Bagging$_{Held-out}$ | 63.13 | 84.41 | 6.16 | 90.97 | 88.24 | 0.97 | 67.55 | 67.27 | 4.82 |
| 120-Bagging$_{Held-out}$ | 63.03 | 82.55 | 7.96 | 91.03 | 88.76 | 1.02 | 68.95 | 69.08 | 5.64 |

In bagging each bag samples N data points with repetition from a dataset of size N. We explore two alternatives, one is sampling 120 bags and sampling for each bag a single model using cSGHMC (named 120 Bagging), and the other is to sample 10 bags and from each to sample 12 models using cSGHMC (named 10 Bagging). The held-out dataset for distillation is constructed in the same fashion as described in the K-Fold alternative. The key difference is that for each data point, there is a different number of models that did not use it in training, due to the stochastic partitioning of bags.

Comparing these two approaches the K-fold has an advantage since each ensemble member was trained on exactly 90% of the training data while each ensemble member in bagging is trained on $\sim 66\%$. Bagging (with 120 bags) however has better diversity as when we distill we see a larger variety of ensemble model combinations.

Table 4 shows results for the Bayesian ensemble used throughout this paper and the method described in this section. As one can see the scores on the held-out datasets, is a good proxy for the test in terms of calibration, agreement, and accuracy (the lower score is due to only averaging a small number of ensemble models), but this comes in many cases at a price of reduced accuracy of the ensemble on the test set compared to the Bayesian ensemble.

Next, we distill the new ensembles using FED. We show the results of the distilled models using the held-out data, and compare them to our Bayesian ensemble distilled using the mixup dataset. As one can see from our results in Table 5, the Bayesian ensemble distilled using the mixup data is superior. We note that on all ensembles the best distillation results we achieved were by using only mixup, further proving the value of this augmentation scheme. We suspect that distillation works best on the Bayesian ensemble as we have the most diverse examples and labels to distill from.

# 6 Conclusions

In this work, we present FED, a novel and general ensemble distillation approach. We show that this approach can distill ensembles while keeping desirable qualities such as accuracy, calibration, and diversity. This is done by maintaining the Bayesian approach of distribution over functions while mimicking the posterior with one that can be evaluated efficiently. Additionally, we show how the need for an auxiliary distillation dataset can be easily solved by using mix-up augmentation. Finally, our experiments show that FED achieves state-of-the-art results on various benchmarks in the domain of Ensemble distribution distillation.

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
