# A Implementation Details

**Data.** To keep experiments uniform, for all datasets (STL-10, CIFAR-10, and CIFAR-100) we used a train/val/test partitioning. The trainset and validset are constructed by a fixed split of 80%:20%.

**Training details for baselines.** In our experiments we compared FED with four baselines. (1) EnDD [5]. (2) EnDD + AUX [5]. (3) AMT [1]. (4) DM - Diversity-Matters [6]. (5) Hydra [7].
For all baselines we tried different learning rates [0.1, 0.01, 0.001] and batch sizes [32, 64, 100]. For EnDD and EnDD + AUX, we used the same temperature, temperature annealing, and optimizer that was used in the original paper. Training for 200 epochs for CIFAR-10, CIFAR-100, with and without AUX. Training for 400 epochs for STL-10 without AUX, and 200 epochs for STL-10 with AUX.
For AMT, we tried different alphas [1e1, 1e3, 1e5] and kept the rest as the original paper. For DM, due to high memory requirements, we were able to go up to a BatchEnsemble with an ensemble size of 8, while being able to use only batch size of 32. In addition, for this baseline we used a bigger memory GPU, unable to fit the training to our standard 11GB GPU used for the rest of our experiments.
For Hydra, due to high memory requirements, we were able to go up to a virtual ensemble size of 8, while being able to use only batch size of 32. Thus, our distilled model had 120 heads, but in training, at each iteration, we sampled randomly 8 heads and trained them. In addition, for this baseline with CIFAR-100 we used a bigger memory GPU, unable to fit the training to our standard 11GB GPU used for the rest of our experiments. Important to note that the distilled model is very heavy due to its 120 different heads, specifically, 3.95GB compared to 43.67MB Generator model for CIFAR-10.

**The Mixup dataset.** In the procedure of creating a Mixup [8] auxiliary dataset, we used a Beta distribution with $\alpha = 0.2$. In Mixup augmentation, and value $\lambda \in [0, 1]$ is sampled from a Beta distribution. $\lambda$ is the interpolation coefficient $mixup - img = \lambda * img_1 + (1 - \lambda) * img_2$.
For STL-10, CIFAR-10, and CIFAR-100 datasets, we created 100k,150k,300k Mixup auxiliary datasets respectively.

**Optimization of the Generator network.** Generator training is done in the following setting: For the optimizer, Adam [3] optimizer with default parameters. For STL-10, CIFAR-10, and CIFAR-100 the best learning rate is 0.001 in the case of the Mixup auxiliary dataset. For the scheduler, we use MultiStepLR scheduler with milestones in epochs [35, 45, 55, 70, 80]. In each milestone multiplying the learning rate by a factor of 0.33. Memory limitations force us to use a virtual ensemble size equal to 8, which essentially means that in each iteration, we sample 8 probability vectors from the 120 probability vectors the auxiliary dataset contains for each example. We use batch size of 64. The combination of batch size = 64 and virtual ensemble size = 8 determines by the GPU memory capacity. We trained the generator for 200 epochs.
In addition, we set the std of the inserted noise to be a learnable parameter as part of the optimization. A value of 0.1 for std is used for initialization.

We share our code with detailed documentation and usage examples for reproducibility.

**Experiments Environment.** All experiments were done on NVIDIA GeForce RTX 2080 Ti with 11019MiB. For Hydra and DM baselines we had to use NVIDIA TITAN RTX with 24220MiB.

# B Additional Experiments

## B.1 Negative log likelihood

We add to our distillation results the negative log-likelihood (NLL) values. Due to space limitations, they were not added to Tab. 2 and appear here 6 instead.

## B.2 Regression on Boston dataset

We evaluated FED performance in a regression task on the Boston dataset from the UCI repository. This dataset has a total of 506 examples, each with 13 features. We allocated 5% from the training set to validation for hyper-parameter tuning. As in the classification tasks we used cSGHMC to obtain samples from the posterior distribution and used FED to distill them. We followed the standard protocol [2] where each model outputs a mean value $\mu(\mathbf{x}; \theta)$ and standard deviation $\sigma(\mathbf{x}; \theta)$ of a Gaussian likelihood for each example. We found that a linear kernel perform best in this setting. We

Table 6: Test accuracy, NLL & ECE (%) ($\pm$ STD) on STL-10 and CIFAR-10/100. Note that the $\text{EnDD}_{AUX}$ baseline uses additional unsupervised data. Each experiment was repeated 3 times.

| | STL-10 | | | CIFAR-10 | | | CIFAR-100 | | |
|---|---|---|---|---|---|---|---|---|---|
| | ACC ($\uparrow$) | NLL ($\downarrow$) | ECE ($\downarrow$) | ACC ($\uparrow$) | NLL ($\downarrow$) | ECE ($\downarrow$) | ACC ($\uparrow$) | NLL | ECE ($\downarrow$) |
| Ensemble | 81.84 | 1.69 | 2.17 | 94.41 | 1.58 | 5.56 | 77.79 | 3.95 | 3.07 |
| $\text{EnDD}_{AUX}$ | **80.78**$\pm$**.00** | 1.71$\pm$.00 | **1.20**$\pm$**.00** | 91.05$\pm$1.3 | **1.55**$\pm$**.00** | 5.63$\pm$1.0 | 70.68$\pm$.00 | **3.92**$\pm$**.00** | 21.40$\pm$.00 |
| EnDD | 73.45$\pm$1.1 | 1.88$\pm$.02 | 10.62$\pm$1.2 | 91.01$\pm$.70 | **1.55**$\pm$**.00** | 6.39$\pm$.81 | 71.16$\pm$.00 | 3.93$\pm$.23 | 14.51$\pm$.00 |
| $\text{DM}_{Gaussian}$ | 74.21$\pm$.13 | 1.89$\pm$.00 | 15.17$\pm$.58 | 90.27$\pm$1.5 | 1.58$\pm$.02 | 2.48$\pm$.94 | 72.44$\pm$.19 | 4.17$\pm$.36 | 4.63$\pm$1.0 |
| $\text{DM}_{ODS}$ | 73.90$\pm$.65 | 1.89$\pm$.00 | 15.32$\pm$.88 | 89.35$\pm$2.7 | 1.59$\pm$.03 | 2.65$\pm$1.2 | 71.85$\pm$.36 | 4.00$\pm$.05 | 4.47$\pm$1.0 |
| $\text{DM}_{ConfODS}$ | 73.26$\pm$2.2 | 1.89$\pm$.00 | 14.52$\pm$3.1 | 89.55$\pm$2.3 | 1.59$\pm$.03 | 2.59$\pm$.29 | 72.33$\pm$.54 | 3.69$\pm$.01 | 5.33$\pm$1.1 |
| $\text{AMT}_{MMD}$ | 54.36$\pm$.78 | 2.26$\pm$.01 | 38.97$\pm$.68 | 83.81$\pm$.56 | 2.21$\pm$.00 | 64.00$\pm$.63 | N/A$\pm$N/A | N/A$\pm$N/A | N/A$\pm$N/A |
| Hydra | 75.74$\pm$1.6 | 2.01$\pm$.02 | 30.68$\pm$3.3 | 91.07$\pm$1.8 | 1.64$\pm$.02 | 5.39$\pm$.61 | 69.56$\pm$.11 | 4.27$\pm$.01 | 29.84$\pm$1.0 |
| FED (Ours) | **80.61** $\pm$ **.36** | **1.69** $\pm$ **.00** | **3.41** $\pm$ **.94** | **93.61** $\pm$ **.06** | **1.55** $\pm$ **.00** | **0.66** $\pm$ **.10** | **74.48** $\pm$ **.10** | 3.95$\pm$ .00 | **1.64** $\pm$ **.17** |

used a Gaussian estimator for the predictive distribution based on sample statistics as in [4, 2]. We found that the NLL of the ensemble was $-0.03$, the NLL of the generator $0.15$, and the average NLL of individual models in the ensemble was $0.78$.

## B.3 Ensemble with fewer models

Table 7: Test accuracy, aggreement, NLL & ECE (%) ($\pm$ STD) on STL-10 with 8 models in the ensemble. Each experiment was repeated 3 times.

| | ACC ($\uparrow$) | AGREEMENT | ECE ($\downarrow$) | NLL ($\downarrow$) |
|---|---|---|---|---|
| Ensemble | 77.05 | 92.96 | 2.61 | 1.77 |
| EnDD | 68.52$\pm$1.4 | 99.99$\pm$.00 | **2.41**$\pm$**.37** | 1.85$\pm$.01 |
| $\text{DM}_{Gaussian}$ | 74.21$\pm$.14 | 86.62$\pm$3.5 | 15.17$\pm$.58 | 1.88$\pm$.00 |
| $\text{DM}_{ODS}$ | 73.90$\pm$0.7 | 86.72$\pm$4.3 | 15.32$\pm$.88 | 1.89$\pm$.00 |
| $\text{DM}_{ConfODS}$ | 73.26$\pm$2.2 | 86.67$\pm$2.7 | 14.52$\pm$3.1 | 1.89$\pm$.01 |
| $\text{AMT}_{MMD}$ | 54.36$\pm$.78 | 11.31 $\pm$.08 | 38.97$\pm$ .68 | 2.26$\pm$.01 |
| Hydra | 72.43$\pm$.91 | 93.26$\pm$.51 | 17.50$\pm$1.2 | **1.75**$\pm$**.01** |
| FED (Ours) | **76.00**$\pm$**.77** | 97.82$\pm$.15 | 8.85$\pm$1.4 | 1.82$\pm$.00 |

In our study, we showed that FED works well with a large number of ensemble models (i.e., 120). We considered this setting since in these cases the inference cost is most debilitating. However, standard studies on Bayesian methods typically sample fewer models from the posterior and one may ask what is the gain in having many models and how would FED perform in this scenario. Figure 4 shows the accuracy, NLL and ECE as a function of the number of models sampled from cSGHMC posterior (up to 40 models). From the figure, the number of models affects mostly on the accuracy but less so on the ECE and NLL. Nevertheless, it shows that there is gain in having more than a few models as typically done. Next, we evaluated FED performance against baseline methods with only 8 models in Table 7. First, the table shows that the performance of all methods degrades compared to having 120 ensemble models. Second, FED achieves the best accuracy and is second in all other performance metrics across all methods. Note that although Hydra is better than FED in terms of agreement and NLL the computational complexity of it (both in memory and run-time) is significantly larger than that of FED. We present a comparison of the run-time for all methods in Section C.1.

## B.4 Different auxiliary datasets

As part of our research to find a suitable dataset for the distillation task, we compared several options. (1) Trainset - the dataset originally used for the ensemble training. (2) Validset - the held-out set during ensemble training, and now can be viewed as new unseen data (without gathering more unlabeled data). (3) Mixup - applying mixup augmentation on the trainset. We were guided by two important aspects. First, the data for the distillation task have to come from a source that does not rely on external or additional data. Second, the data needs to be easily created/generated.

### B.4.1 Extended ensemble analysis

In table Table 8 we present an extended ensemble analysis of Accuracy, agreement, and calibration(ECE) over the different ensembles and different datasets.

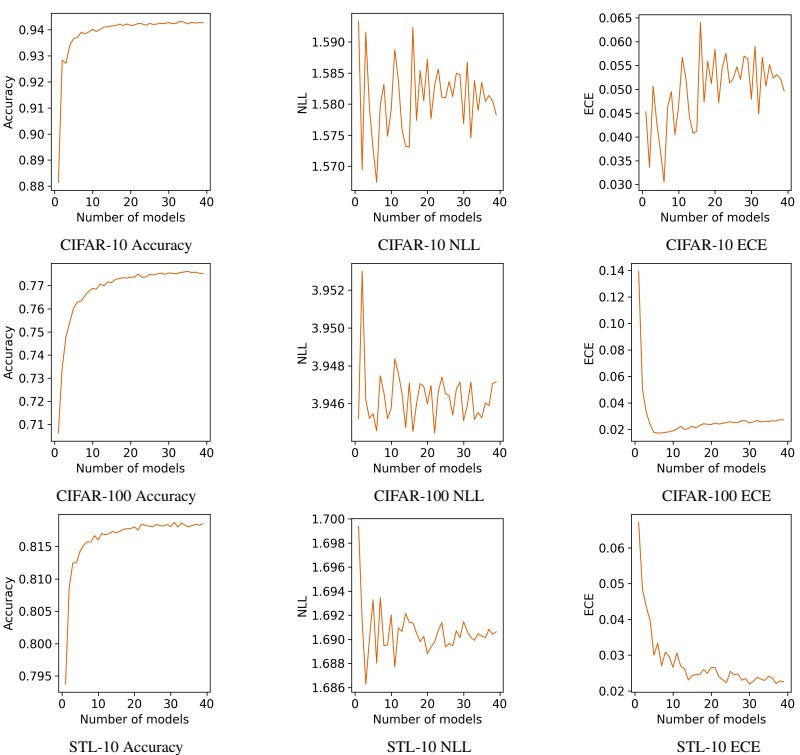

Figure 4: Accuracy, NLL and ECE of the Bayesian ensemble as a function of the number of models sampled from the posterior.

Table 8: Ensemble analysis. Test accuracy, agreement & ECE (%) on STL-10, and CIFAR-10/100.

| | STL-10 | | | CIFAR-10 | | | CIFAR-100 | | |
|---|---|---|---|---|---|---|---|---|---|
| | ACC (↑) | AGREEMENT | ECE (↓) | ACC (↑) | AGREEMENT | ECE (↓) | ACC (↑) | AGREEMENT | ECE (↓) |
| Train | 100.0 | 98.23 | 2.99 | 99.97 | 91.85 | 6.38 | 99.95 | 97.53 | 4.15 |
| Valid | 81.20 | 87.77 | 3.17 | 94.91 | 86.08 | 6.18 | 78.31 | 70.67 | 4.38 |
| Test | 81.84 | 87.85 | 2.17 | 94.41 | 86.08 | 5.56 | 77.79 | 71.97 | 3.07 |
| mixup | – | 95.20 | – | – | 88.08 | – | – | 88.56 | – |

### B.4.2 Generator with different auxiliary datasets

In the following Tables 9, 10, 11, and 12 we show results of training a Generator with different auxiliary datasets, over different ensembles - Bayesian, 10-Fold, 10-Bagging, and 120-Bagging respectively. In the Bayesian ensemble, the validset is a held-out set (20% of the entire dataset). For STL-10, CIFAR-10, and CIFAR-100 the validset is of size 1k, 10k, and 10k respectively. For 10-Fold, 10-Bagging, and 120-Bagging the held-out set is constructed as the entire trainset. But, each example uses for prediction only models in the ensemble that did not use this example in their training. This means that for the three ensemble alternatives, the held-out set is essentially the entire trainset using subsets of the ensemble. For STL-10, CIFAR-10, and CIFAR-100 the held-out size is 5k, 50k, and 50k respectively.

Results in Table 9 show that using our Mixup augmentation for auxiliary dataset yields the best accuracy and calibration. One observation is that the agreement of the generator trained with the validset is better. This is no surprise, the validset is essentially new unlabeled data that comes from the same distribution as the testset. While being better in terms of the agreement, the accuracy substantially decreases due to the small size of the validest. Thus, our main effort was indeed to find a method for creating more data that resembles the testset distribution, and we found Mixup as a good option.

On 10-Fold 10, 10-Bagging 11, and 120-Bagging 12 the results share the same insights as on the Bayesian ensemble. In addition, thanks to the high diversity that comes from the ensemble itself,

Table 9: Ablation - FED with different auxiliary datasets on **Bayesian Ensemble**. Test accuracy, agreement & ECE (%) on STL-10, CIFAR-10, and CIFAR-100.

| | STL-10 | | | CIFAR-10 | | | CIFAR-100 | | |
|---|---|---|---|---|---|---|---|---|---|
| | ACC (↑) | AGREEMENT | ECE (↓) | ACC (↑) | AGREEMENT | ECE (↓) | ACC (↑) | AGREEMENT | ECE (↓) |
| Trainset | 65.06 | 97.18 | 5.00 | 92.58 | 99.16 | 1.92 | 68.34 | 97.58 | 9.61 |
| Validset | 53.31 | 94.07 | 6.30 | 82.64 | 83.60 | 3.47 | 54.13 | 87.17 | **1.43** |
| Mixup | **80.94** | 97.16 | **2.55** | **93.68** | 93.71 | **0.78** | **74.56** | 88.20 | 2.07 |

the distilled model also has a high diversity. In terms of accuracy, Mixup still has the upper hand. Furthermore, the held-out has a better agreement and better accuracy compared to the validset used in the Bayesian ensemble. A direction worth exploring in our opinion is to apply Mixup augmentation over the held-out set, thus, enjoying a large amount of unseen data.

Table 10: Ablation - FED with different auxiliary datasets on **10-Fold Ensemble**. Test accuracy, agreement & ECE (%) on STL-10, CIFAR-10, and CIFAR-100.

| | STL-10 | | | CIFAR-10 | | | CIFAR-100 | | |
|---|---|---|---|---|---|---|---|---|---|
| | ACC (↑) | AGREEMENT | ECE (↓) | ACC (↑) | AGREEMENT | ECE (↓) | ACC (↑) | AGREEMENT | ECE (↓) |
| Trainset | 64.51 | 97.65 | 8.75 | 91.04 | 94.73 | 4.82 | 66.43 | 86.43 | 10.72 |
| Held-out | 62.92 | 84.40 | **5.33** | 91.00 | 90.48 | **0.87** | 67.73 | 80.39 | 1.64 |
| Mixup | **77.32** | 97.25 | 9.65 | **93.84** | 94.10 | 0.99 | **74.99** | 84.51 | **1.62** |

Table 11: Ablation - FED with different auxiliary datasets on **10-Bagging Ensemble**. Test accuracy, agreement & ECE (%) on STL-10, CIFAR-10, and CIFAR-100.

| | STL-10 | | | CIFAR-10 | | | CIFAR-100 | | |
|---|---|---|---|---|---|---|---|---|---|
| | ACC (↑) | AGREEMENT | ECE (↓) | ACC (↑) | AGREEMENT | ECE (↓) | ACC (↑) | AGREEMENT | ECE (↓) |
| Trainset | 66.21 | 74.97 | 6.95 | 91.68 | 90.69 | 1.54 | 68.53 | 76.06 | 2.61 |
| Held-out | 63.13 | 84.41 | **6.16** | 90.97 | 88.24 | **0.97** | 67.55 | 67.27 | 4.82 |
| Mixup | **77.37** | 95.47 | 10.24 | **93.76** | 90.43 | 1.52 | **73.73** | 68.95 | **4.68** |

Table 12: Ablation - FED with different auxiliary datasets on **120-Bagging Ensemble**. Test accuracy, agreement & ECE (%) on STL-10, CIFAR-10, and CIFAR-100.

| | STL-10 | | | CIFAR-10 | | | CIFAR-100 | | |
|---|---|---|---|---|---|---|---|---|---|
| | ACC (↑) | AGREEMENT | ECE (↓) | ACC (↑) | AGREEMENT | ECE (↓) | ACC (↑) | AGREEMENT | ECE (↓) |
| Trainset | 63.50 | 76.03 | 8.91 | 91.26 | 90.92 | 2.15 | 69.12 | 76.88 | **2.95** |
| Held-out | 63.03 | 82.55 | **7.96** | 91.03 | 88.76 | 1.02 | 68.95 | 69.08 | 5.64 |
| Mixup | **76.34** | 95.64 | 13.91 | **93.54** | 90.84 | **0.97** | **74.26** | 72.53 | 3.22 |

## C  Performance

### C.1  Runtime

We show in Figure 5 that for a wide range of batch sizes the run-time of our model scales approximately as $1 + (\# \ of \ \epsilon \ samples)/35$ resulting in a considerable speedup. Specifically, the generator inference time with 120 epsilons takes 0.020 seconds, while ensemble inference time takes 4.075 seconds. Also the figure shows a comparison of the accuracy between FED and baseline methods as a function of the inference time. With only a few $\epsilon$ samples FED outperforms all baseline methods both in accuracy and run-time efficiency. Adding more $\epsilon$ samples further improves the models performance at the expanse of a slight degradation in the inference time.

### C.2  Memory

For STL-10 and CIFAR-10, the memory requirement of the entire ensemble is 5.117GB, i.e 120 models of ResNet-18 that each requires 43.669MB.
Our generator, which essentially is almost a regular ResNet-18, requires 43.677MB.

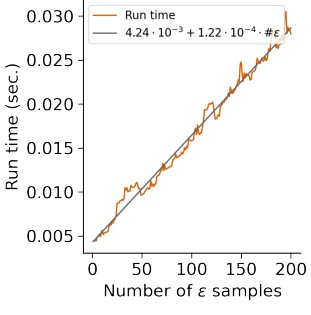

(a) FED Inference Time

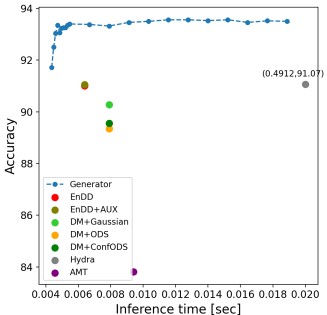

(b) CIFAR-10 Inference Time Comparison

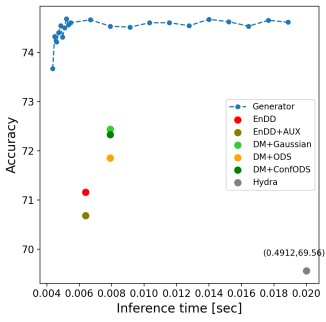

(c) CIFAR-100 Inference Time Comparison

Figure 5: (a) FED inference time as a function of the number of $\epsilon$ samples. (b) and (c) accuracy vs inference time for all compared methods. FED was evaluated with 1, 2, ,.., 10, 20, ..., 120 $\epsilon$ samples.

For CIFAR-100, the memory requirement of the entire ensemble is 5.261GB, i.e 120 models of ResNet-18 that each requires 43.849MB.
Our generator, which essentially is almost a regular ResNet-18, requires 43.858MB.

# D   Dirichlet goodness of fit

Table 13: Dirichlet distribution fitness test.

|  | STL-10 AGREEMENT | CIFAR-10 AGREEMENT | CIFAR-100 AGREEMENT |
|---|---|---|---|
| Ensemble | 87.85 | 86.08 | 71.97 |
| Dirichlet | 91.01 | 90.66 | 77.06 |

Few of our baselines consider the task of distillation as learning the Dirichlet distribution over the prediction for every input example. We argue that the Dirichlet distribution imposes a limitation on the expressive power of the distilled model. Thus, introduces an irreducible error in the distilled model. In order to show that, we test how well the Dirichlet distribution fits the probability vectors that the ensemble produces.

In the following experiment, we take each example from the testset, and perform an MLE procedure to find the Dirichlet distribution that most likely produces those probability vectors $\{p(y_*|\mathbf{x}_*, \theta_1), ..., p(y_*|\mathbf{x}_*, \theta_M)\}$. Next, we test the Agreement metric introduced in Section 5 over probability vectors sampled from the computed Dirichlet distribution, i.e $\{y_i\}_{i=1}^{M} \sim Dirichlet(alphas)$. Suppose the Dirichlet distribution expressiveness is good enough, we would expect a perfect match to the predictive distribution of the ensemble on this example.

Results in Table 13 suggest that indeed, there is a gap between the predictive distribution of the ensemble and the maximum likelihood Dirichlet distribution in terms of the agreement. For all three datasets, the agreement of the ensemble over the testset is lower (higher diversity) than the agreement of the Dirichlet distribution over the testset. This means, that leaning on a Dirichlet-based distilled model has a fundamental limitation.