# OpenReview forum: "Functional Ensemble Distillation"
_NeurIPS.cc/2022/Conference — NeurIPS 2022 Accept_

### Official Review · Reviewer_rtLD · 2022-06-28

**Rating:** 7
**Confidence:** 4
**Soundness:** 3 good
**Presentation:** 4 excellent
**Contribution:** 4 excellent

**Summary:**

The authors propose a method for distilling an ensemble of models. This improves performance (only one model must be stored and used). Rather than making any substantial assumptions about the probabilistic model, the method uses relatively small amounts of added noise and data augmentation.

**Questions:**

Mostly discussed above:
- Why noise concat and addition?
- What actually is the overhead?
- Augmentation ablation?
- Why so many ensemble elements? Does it break with fewer?
- Is there evidence for the claim that ensemble agreement hurts the distilled model?
- What is the 'Bayesian' baseline compared to the 'Ensemble' baseline?

**Limitations:**

Seems like there might be an ensemble element number limitation that isn't addressed?

**Strengths And Weaknesses:**

I liked this paper. It is clear and very well written. I expect it to have significant impact and it is fairly original. Most of my comments are minor and reflect things I feel could have made the arguments even stronger.

The title/name is 'Functional' but the method doesn't feel super functional to me. Part of the loss is arguably functional, but the method is basically parametric. Is there a clearer way to communicate the method?

ln 39 "using" and "use" redundant.

There was some overlap of background/intro/related work. Perhaps a little more editing could let you hit your contribution sooner.

I don't understand the motivation for concating 3 Gaussian RVs to the input and adding noise (is it a vector? scalar?) for each block. You mention you considered other things in teh paragraph starting ln 177. Are you able to provide ablations? It isn't a deal breaker, but this felt pretty arbitrary.

In line 184 you mention non-negligible overhead. Please can you just say what the overhead is? In params and FLOPS/ms?

You mention in ln 205 that you tried a GAN thing and it didn't work as well. Can you just provide the ablation in the appendix? There's a general theme here that you allude to some work investigating alternatives and it feels like you could maybe just show the work? This could go into Table 3.

In line 234, a nitpick. It's not that I think you *should* use batchnorm, but I think you are overstating the point. First, Ismailov et al. only say that batch norm makes it harder to clearly interpret the probabilistic model, not that it is wrong. I think that's the correct view---the probabilistic model becomes a bit more complicated but it isn't all of a sudden non-Bayesian. Second, Bayesian inference doesn't make an IID assumption---in fact one of the most important parts of the Bayesian approach is that the data are not distributed at all! So I think this statement is incorrect.

You use 120 models for your ensemble. That's rather a lot. Typically people doing ensembling might use between 3 and 10. Is there a reason for your choice? Is it perhaps important for your method to have lots of ensemble elements? In that case there might be an unacknowledged limitation of the approach, that during training you have to produce many more ensemble elements in order to get a good distillation (so you are trading off training time for test time). I would like to see some ablations with 5 ensemble members, say, to understand this tradeoff better.

Regarding ensemble agreement, you do show in Table 1 that the ensemble members have high agreement on the training data. But you claim that this leads to problems with the distilled ensemble. I didn't understand you to have provided any evidence for that second claim. Have I missed it?

In Table 2 there are many apparent errors in what has been made bold. For example, FED is shown bold for ECE on STL despite the fact that it is definitely not as good as EnDD_{AUX} (I understand of course that it doesn't see as much data!). Similarly, in some sense AMT has the 'best' agreement on STL-10 but it is DM_G which is bolded (and moreover the three DM methods are statistically indistinguishable). If you are going to bold things, please can you go through this carefully to make sure that the bold is correct (I don't claim to have caught all the corrections).

The 'Bayesian' baseline in Table 4 looks like it has overfit quite badly. Do you think that maybe there is a tuning problem and the baseline is a bit weak? Also, from the body of the text it seems like 'Bayesian' is the same as 'Ensemble' from other tables? Or is there a reason you use a different name?

---

> ### Author Response · Authors · 2022-08-01
> **Response to rtLD**
>
> We thank you for your comments, please see the general comment for the response to commonly raised questions. We fixed the corrections to the writing you pointed out.
>
> - “I don't understand the motivation for concating 3 Gaussian RVs to the input and adding noise (is it a vector? scalar?)” - Thank you for this comment. We will clarify this part. First, the noise is indeed a vector. We needed to inject enough noise to capture the diversity in the ensemble models. Second, initial experiments showed that this design choice works better compared to other simpler alternatives (such as summation of the noise). Since it felt like a minor point we did not add it to the ablation study.
>
> - “In line 184 you mention non-negligible overhead. Please can you just say what the overhead is?” - FED runtime depends on the number of $\epsilon$ samples we take. The run-time increases approximately like with the number of $\epsilon$s divided by 35. To measure this effect we show in supp. section C FED run time as a function of the number of $\epsilon$s and compare FED with baseline methods in terms of accuracy and run-time. The comparison shows that with only a few $\epsilon$s sampled, FED is faster and more accurate than any of the alternatives. If we sample many $\epsilon$s, there is a slight increase in the performance in return of an increase in the run-time. Finally, we believe that the number of FLOPs isn’t a preferred metric for quantifying the overhead in this case as we can parallelise computations to get an overall quicker run-time.
>
> - “You mention in ln 205 that you tried a GAN thing and it didn't work as well. Can you just provide the ablation in the appendix? There's a general theme here that you allude to some work investigating alternatives” - In this work, like many others, we investigated multiple approaches before settling on the direction proposed in this paper. We feel that it is important for the community if we do not just present what worked, but also mention directions that did not pan out. However, the rigor required from the results presented in the paper is higher than for initial investigation so adding these experiments will add considerable compute and time requirements.
>
> - Augmentation ablation: The OOB experiment was our attempt to do something in that nature.
>
> - Is there evidence for the claim that ensemble agreement hurts the distilled model? I would say that ensemble agreement can hurt when it is very high, as it is not too far from simply training on the original labels. We have some evidence, as STL10 and CIFAR100 that had very high agreement showed considerable improvement from mix-up compared to distilling on the training data. On CIFAR10 where the agreement was not as high, the gains were marginal.
>
> - Bayesian baseline: First we fixed the name in table 4 to be consistent. In Bayesian inference, every single prediction can overfit to the training data but we get good generalization by averaging over multiple predictors. This is why very high accuracy on the training set is to be expected and is not an issue.
>
> - Batchnorm: Made a small change to the working to say the i.i.d assumption is only commonly used.

---

> > ### Comment · Reviewer_rtLD · 2022-08-07
> > **Thanks for your response**
> >
> > Your points seem reasonable. I don't think they changed my view particularly, but I was already recommending acceptance so that's not a problem.
> >
> > Edit: I see that you answered my question about ensemble size in the top-level response.

---

> > > ### Author Response · Authors · 2022-08-10
> > > **Thank you**
> > >
> > > We thank the reviewer for the valuable feedback and score.

---

### Official Review · Reviewer_i7D4 · 2022-07-06

**Rating:** 6
**Confidence:** 4
**Soundness:** 3 good
**Presentation:** 3 good
**Contribution:** 3 good

**Summary:**

This paper proposes Functional Ensemble Distillation (FED), a new method for distilling (the predictions of) an ensemble of models into a single model. FED trains a generator network which learns a distribution over functions by mapping the input together with a sample of an external random noise variable to the output. As forward passes with the distilled generator model can be made for multiple samples of the random noise variable in parallel, the method is efficient to run at test time. In contrast to previous work, their approach can also quantify the correlations between predictions. Furthermore, the paper provides a solution to the case where all models in the ensemble overfit to the training data and make very similar predictions to achieve zero training loss, such that distilling their predictions becomes sub-optimal: using an auxiliary dataset generated via mixup data augmentation, the individual model predictions can be made much more diverse, which helps distill a more performant model. The paper empirically validates the efficacy of the proposed distillation method in terms of predictive accuracy and calibration compared to other baselines.

**Questions:**

- Do you have some intuition as for why the OOB methods in Section 5.4 perform substantially worse than the mixup approach? Is it because they simply use less data for training? Or is it because mixup introduces additional regularization that boosts performance?
- I'm not sure if I fully understand what you mean when saying that FED scales approximately as 1 + (# of epsilon samples) / 35; e.g., for 120 epsilon samples, this would roughly be 1 + 120 / 35 = 4.43; is this referring to a multiplicative factor relative to the inference time of a single deterministic model (i.e. saying that FED with 120 samples is ~4.5x slower than a single model)?
- Relatedly, in Appendix C.1 you mention that inference with an ensemble of 120 models takes 4.075s wall-clock time. As inference time should grow linearly with the number of ensemble members, I would expect a single model to roughly take 4.075s / 120 = 0.034s for inference. It is then claimed that FED takes only 0.020s with 120 epsilons, which is almost twice as fast as that. How can it be that FED inference time is faster than that of a single deterministic model? Following your claim of the approximate scaling of FED, shouldn't the runtime of FED be much higher than what you report? Or am I misunderstanding something here?
- Why did you not include the most basic distlliation baseline (i.e. simply the original Hinton et al. distillation procedure) in your evaluation? I believe this would be insightful to some readers.


**Limitations:**

Yes, the authors have adequately addressed this.

**Strengths And Weaknesses:**

**Summary of Review**

The proposed new method to address the important problem of ensemble distillation is clever and empirically shown to significantly boost performance in some settings. While the method seems promising, I see some issues with the experiments (detailed below) that should be addressed to strengthen the work. Overall, I am leaning towards recommending rejection of the paper in its current form, but I am happy to reconsider my judgement if my concerns can be adequately addressed.

**Strengths**
- The manuscript is overall well written and easy to follow.
- The work tackles an important problem, namely making ensembles more efficient via distillation.
- The proposed FED approach for distilling an ensemble using a generator network with an external noise random variable is novel, sound, and intuitively sensible.
- The proposed method for diversifying the ensemble member predictions in the overfitting regime by generating an auxiliary dataset via mixup augmentation is simple and intuitively appealing.
- The empirical evaluation is thorough and demonstrates that FED can achieve significant performance gains in terms of predictive accuracy and calibration compared to relevant baselines across several image classification benchmarks.

**Weaknesses**
- I am not sure if the empirical comparison is fair, as the different methods seem to have different runtime requirements; in particular, you claim that the inference runtime of FED scales as 1 + (# of epsilon samples) / 35, which for 120 samples (i.e. the number you use in the experiments) is roughly 4.5; in my understanding, this means that the inference time of FED is 4.5x slower than that of a single deterministic model (while in contrast, the vanilla ensemble is 120x slower; I requested clarification for this in the Questions below, so please let me know if my understanding is incorrect); if this assumption is correct, this would mean that FED is significantly slower than some of the baselines considered; e.g., in my understanding, EnDD should have a runtime similar to that of a single model (again, please correct me if I'm mistaken); all in all, my concern is that the performance comparison should be conducted more carefully, taking inference runtime into account, to make it more meaningful; to this end, one could e.g. plot the results in 2D with runtime on the x-axis and performance on the y-axis; different methods would then result in different trade-offs on this plot, yielding Pareto curves that would be much more insightful than just single performance numbers; at the end of the day, a practitioner is probably interested in one of two questions: 1) given a fixed runtime budget X, what’s the best performance Y that I can achieve?, or 2) given a desired performance Y, what’s the fastest runtime X to achieve it? to make the comparison even more interesting, it would be great to consider different settings for each individual method for trading off between runtime and performance; I'm not fully sure how to best do this, but e.g. for FED, one could try varying the number of epsilon samples (for a fixed number of ensemble members) to make the method cheaper; conversely, one could try to make the baselines more expensive in more way; this would allow you to compare performance of the methods for a given fixed inference budget.
- The empirical evaluation could be significantly improved by considering a more diverse set of model architectures and benchmarks; the paper focuses on a single architecture (ResNet18) and three small-to-medium-sized image classification benchmarks (CIFAR-10, CIFAR-100, STL-10); it is not clear if the drawn conclusions would extend to other model families (e.g. transformers), data modalities (e.g. text) and benchmark sizes (e.g. ImageNet); to gain more space for reporting additional major empirical results, one could e.g. move Section 5.4 (on the OOB methods for inducing diversity without an auxiliary dataset) to the appendix -- while the experiment and conclusion in that section are certainly interesting, they do not seem to necessary to have in the main text of the paper due to the comparably poor results.
- The paper emphasises that, in contrast to previous methods (which are e.g. based on fitting a Dirichlet distribution over predictions), FED is able to capture correlations between outputs, and claims that this is an important feature for many real-world applications (e.g. l. 9-11, l. 52-55, l.149-153); as this is mentioned already prominently in the abstract as a main selling point for the method (l. 9-11), I was expecting at least some empirical evidence for the benefit of this feature; unfortunately, no explicit evidence is provided beyond some vague intuitive written arguments (l. 52-55, l. 149-153). It would significantly strengthen the paper if capturing these covariances could be shown to improve performance in some application. (While I understand that FED is shown to improve upon previous methods, it is not clear to me how this gain can be directly attributed to the aforementioned property.)

**Minor Issues**
- Table 3: bolding the best values would improve clarity
- It would be useful to put a brief take-away message into the caption of each table/figure
- The paper misses the Laplace approximation as another fundamental Bayesian inference approach for neural networks (in addition to VI and MCMC); for a recent overview, see e.g. Daxberger et al., *Laplace Redux -- Effortless Bayesian Deep Learning*, NeurIPS 2021
- In addition to the runtime comparison requested above, it would also be useful to have a more clear/complete comparison of the memory requirements of all methods; you mention the memory burden of FED in Appendix C.2, and e.g. that of the Hydra baseline in Appendix A; it would be useful to have a table/plot comparing this for all methods considered

---

> ### Author Response · Authors · 2022-08-01
> **Response to i7D4**
>
> We thank you for your comments, please see the general comment for the response to commonly raised questions.
>
>
> - “I am not sure if the empirical comparison is fair, as the different methods seem to have different runtime requirements;”  Thank you for the detailed comment. Following your suggestion, we added a scatter plot that shows the inference time vs the accuracy in Supp. section C.1. The figure shows that with only a few $\epsilon$'s sampled, FED is faster and more accurate than any of the alternatives. If we sample many $\epsilon$'s, there is a slight increase in the performance in return for an increase in the run-time.
>
> - “The empirical evaluation could be significantly improved by considering a more diverse set of model architectures and benchmarks” - Thank you for the suggestions. Regarding larger datasets (e.g., Imagenet) please see our comment in the general remarks. Also, following this comment, and due to lack of time, we added another experiment solving a regression task with a fully connected network. Results and further details are now in supp. section B.2.
>
> - “Do you have some intuition as for why the OOB methods in Section 5.4 perform substantially worse than the mixup approach?” - OOB methods have a tradeoff: If each model is trained on small amounts of data then every ensemble element is worse, if they are trained on most of the data then we have little OOB diversity for training the generator. We note that their subpar performance surprised us as well.
>
> - “you mention that inference with an ensemble of 120 models takes 4.075s wall-clock time. As inference time should grow linearly with the number of ensemble members, I would expect a single model to roughly take 4.075s / 120 = 0.034s for inference” - Running the 120 model ensemble is slower as they do not all fit in memory which leads to extra time spent on loading the models.
>
> - Naive baseline - Thank you for this comment. The naive baseline was added to Tab. 2&3. We note that on cifar10 and cifar100 it is better or equal to all other distillation methods except our FED approach.
>
> - VI/Laplace approximation - We do talk about variational inference in general but the focus of this work is to speed up Monte-Carlo samples, so VI or the Laplace approximation is out of the scope of this paper. We will add the reference to the related work.

---

> > ### Comment · Reviewer_i7D4 · 2022-08-08
> > **Thank you for your response**
> >
> > Thanks a lot for your thorough and helpful response and the additional results, which resolves most of the concerns I had. I am particularly impressed by the runtime vs. performance results for different numbers of epsilon samples, which clearly show that the proposed method is superior in addressing this trade-off than the considered baselines. After also having read all other reviews and the corresponding author responses, I am therefore increasing my score from 4 to 6 and am now thus inclined to recommend acceptance. The remaining concern of lack of larger-scale experiments makes me hesitate to make my recommendation more enthusiastic.

---

> > > ### Author Response · Authors · 2022-08-10
> > > **Thank you**
> > >
> > > We thank the reviewer for reassessing the paper and raising the score based on our response.

---

### Official Review · Reviewer_a1xo · 2022-07-12

**Rating:** 6
**Confidence:** 4
**Soundness:** 3 good
**Presentation:** 3 good
**Contribution:** 3 good

**Summary:**

This paper proposes a novel approach for distilling a Bayesian ensemble of models:

$\int_{\theta}p(y | x, \theta) p(\theta | D) d\theta \approx \frac{1}{M}\sum_{\theta_i \sim p(\theta | D) } p(y | x, \theta_i) $

Existing methods tries to distill the ensemble by fitting a Dirichlet distribution:

$f(x; \phi) = \alpha$, where $\alpha$ is the concentration parameters that fully characterize the Dirichlet distribution

This paper proposes to learn a generator that can generate predictive distribution like
$p(y | x, \epsilon_i)$, where $\epsilon_i$ are random samples from a fixed distribution.

This is in spirit very much like the reparameterization trick for variational auto-encoder [1].

The inference can be executed by sampling a batch of $\epsilon$’s and then run the forward function of the learned generator on this batch in parallel.

To promote the diversity of the ensemble, during training the generator, mixup data augmentation is applied, which is shown to significantly boost the accuracy and diversity.


[1] Kingma & Welling, Auto-Encoding Variational Bayes, 2013


**Questions:**

- how does it compare, in terms of accuracy, to a naive baseline where we use multiple teachers (i.e., the ensemble of models) to distill one student model $P(y|x)$?


- What's the decision making algorithm given an ensemble of predictions or a Dirichlet distribution? Averaging $p(y | x, \theta_i)$ or majority vote? for EnDD, do you take the mean of the learned Dirichlet distribution or take samples from that and then take majority vote? If the former, how do you measure agreement metric?

- could you elaborate on how to set the noise STD to be a learnable parameter?


**Strengths And Weaknesses:**

Strengths

- The proposed method is novel and general purpose, can distill any ensemble of models
- The idea of using mixup augmentation for distillation is creative and interesting
- The experimental results seem strong, with well designed ablation studies on mixup.

Weaknesses:

- it’s claimed to be a general-purpose distillation method, but only classification tasks are considered in the experiments; it’s not clear how good it performs on regression tasks, and maybe compare with EnDD fitting, e.g., a Gaussian distribution (which seems straightforward to adapt)
- one advantage of the proposed method is claimed to be able to measure the correlation between two inputs, but seems it’s not discussed how to do that, and not experimentally validated that the correlation is a close approximation of the original ensemble
- there seems to be large computational overhead during inference since each example need to be extended to a batch with different $\epsilon$ samples, so inference cost is multiplied by the number of samples, though they could be run in parallel.

---

> ### Author Response · Authors · 2022-08-01
> **Response to a1xo**
>
> We thank you for your comments, please see the general comment for the response to commonly raised questions.
>
> - “Only classification tasks are considered” - We focused on classification as this is the main task used in previous studies and on which we can compare our method to baseline models. Following this comment, we ran an experiment on the Boston regression dataset from the UCI repository (supp. Sec. B.2) which showed that FED distills well in this setting also. Unfortunately, due to lack of time, we were not able to compare FED with relevant baseline methods (e.g., Hydra or EnDD).
> - “there seems to be large computational overhead … though they could be run in parallel.” Being able to run fast in parallel is a key part of our model and a main reason it was designed as such. Supp Sec. C shows the growth of FED run-time as a function of the number of smapled $\epsilon$. Following this comment, we also compare FED with baseline methods in terms of run-time and accuracy in Supp Sec. C. With only a few $\epsilon$s sampled, FED is faster and more accurate than any of the alternatives.
> - Naive baseline: Thank you for this comment. The naive baseline was added to table 2&3. We note that on cifar10 and cifar100 it is better or equal to all other distillation methods except our FED approach.
> -“What's the decision making algorithm given an ensemble of predictions or a Dirichlet distribution? Averaging $p(y|x,θ_i)$?” Yes. We obtain samples and from them compute the average  $p(y|x,θ_i)$ and agreement.
>
> - How the noise STD is trained: It is a scalar that multiplies the incoming normal noise before it is added. It is optimized like other parameters via backpropagation.
>
> - memory requirements: We discussed memory requirements in Sec. C.2 in the supplementary.

---

### Official Review · Reviewer_m5GE · 2022-07-13

**Rating:** 6
**Confidence:** 4
**Soundness:** 2 fair
**Presentation:** 3 good
**Contribution:** 3 good

**Summary:**

This paper proposes a functional ensemble distillation (FED), a novel approach for distilling an ensemble of deep neural networks into a single model. The main idea is to create a generator in a student network that takes Gaussian noises as additional inputs and emulates samples from the posteriors from them and learns the diverse prediction of ensemble teachers in a nonparametric fashion using Maximum Mean Discrepancy (MMD). Additionally, to resolve the issue of ensemble teachers almost completely agreeing with each other on the training set, the paper further proposes to use mixup augmentation during training. The proposed method is validated on some benchmark image classification tasks.

**Questions:**

- The paper seems to mainly discuss the setting where an ensemble constructed from cSGHMC are used as a teacher. Does it work well with the typical deep ensembles with a much less number of ensemble models, $M \in (4~8)$ for instance?
- Similar to the above question, how does FED perform w.r.t. the different number of ensembles $M$? Does it work well with a much less number of ensemble members? This is particularly important because in real-world scenarios it is often infeasible to use 120 samples during training in a restricted resource setting.
- I find the results confusing in some sense; most of the time, FED shows higher accuracies and lower ECEs than the baselines, but the agreement values are quite bad, which potentially indicates that the predictions produced by FED are not as diverse as teachers. This contradicts the previous reportings that the diversities in the predictions are actually key to the robustness and better uncertainty quantification in ensembles. Can you elaborate on this?
- Usually, when it comes to comparing ECEs, it is recommended to first do the temperature scaling and compare the ECEs with the optimal temperatures. Judging from the numbers in the results, the scales of the ECEs are not similar to each other, so I suspect that the models are not temperature-scaled before computing ECEs. Can you further compare the ECEs with calibrated models?
- In my opinion, ECE is not a good metric on its own to capture various aspects of uncertainty quantification of the model, so it is always beneficial to look at multiple uncertainty metrics such as negative log-likelihoods, brier scores, and out-of-distribution detection performances. Especially, related to the question above, since the proposed model exhibits high accuracy but high agreement values, it would be important to look at these metrics.
- The baselines can also benefit from mixup augmentation. Have you try applying it for the baselines?


**Limitations:**

- As far as I see from the paper, DM and Hydra use only 8 heads but still distill from 120 samples; this might seriously harm the performance of the baselines, especially because these methods are designed to distill each teacher member to each subnetwork in a one-to-one fashion. If possible, it would be good to see the comparison in the setting where the teachers come with a small number of ensemble members so that one can directly distill DM and Hydra in a one-to-one fashion.
- As mentioned above, no metrics other than ACCs and ECEs.
- No experiments on large-scale image classification benchmarks, such as ImageNet. I find the distillation method often suffers when it comes to the scale of ImageNet, so would be good to see the scalability of the proposed methods on such datasets.
- The main contribution can be understood as a combination of two ingredients; 1) design of the student networks that introduce Gaussian perturbations for inputs and intermediate layers, and 2) learning with MMD. Without an ablation study, it is hard to see the net effect of each component. For instance, we can try learning the proposed model with typical knowledge distillation loss, or try distilling a Hydra architecture with MMD loss.

**Strengths And Weaknesses:**

Strengths
- The paper is relatively well written and easy to follow.
- The motivation is well set.
- The distillation via MMD seems novel (in the ensemble distillation literature, but not in general).
- The experimental results seem promising.

Weaknesses
- Only evaluated with accuracy and ECE, so the results do not fully reflect the uncertainty quantification aspect of the models.
- Somewhat unfair comparison environment for DM (see below).
- No experiments on larger scale datasets such as TinyImageNet or ImageNet.

---

> ### Author Response · Authors · 2022-08-01
> **Response to m5GE**
>
> We thank you for your comments, please see the general comment for the response to commonly raised questions.
>
> - “Only evaluated with accuracy and ECE” - In the paper we also show the agreement metric and results on out-of-distribution detection. We added NLL results to the supplementary.
>
> - “the agreement values are quite bad”  While the agreement values aren't as good as those of the original ensemble, they are still better in many cases than the values of the baselines. This is partially due to the fact that the mix-up agreement we train on can be in some cases higher than that of the ensemble. Distilling an ensemble with better agreement was something we focused on and it did lead to better accuracy. There is still a gap in agreement and it will be the focus on further work. We also note that in the distillation as the samples are not i.i.d more diversity does not always mean better results.
>
> - “when it comes to comparing ECEs, it is recommended to first do the temperature scaling” - while ECE can be greatly improved by temperature scaling, that is also true about a single predictor and is orthogonal to the ensemble distillation. We use ECE as another metric to see if we captured the ensemble behavior. Also, temperature scaling requires another validation set which might be unavailable in setups with limited data.
>
> - “The baselines can also benefit from mixup augmentation. Have you try applying it for the baselines?” This was done in table 3 where we used mixup with EnDD and our model was superior.
>
> - “DM and Hydra use only 8 heads but still distill from 120 samples;” It seems that the experimental details in the supplementary material were misleading. Thank you for pointing that out, we will clearify. DM was indeed trained with 8 heads but from 8 ensemble models in a one-to-one fashion because of its high memory requirements. Hydra, on the other hand, was trained with 120 heads, only that we were able to train it with 8 heads at each iteration (due to high memory requirement). Following this comment, we added another experiment (sec. B.3 in the appendix) in which all methods distilled 8 ensemble models on the STL-10 dataset. We show in this case as well that FED has the best accuracy (in addition to a low memory footprint compared to Hydra).
>
> - “The main contribution can be understood as a combination of two ingredients…..Without an ablation study, it is hard to see the net effect of each component.” First, we note that using mixup for distillation is another contribution. Second, these two parts are connected, as we treat the distill model as a generative model we need an appropriate loss to train a generative model.

---

> > ### Comment · Reviewer_m5GE · 2022-08-07
> > **Thanks for the clarification**
> >
> > I appreciate the additional results which clarified some of my concerns, so raise my score to 6.
> > Still, since DM and Hydra are not designed to distill from the samples from SGHMC, but distinct modes obtained from deep ensembles, I wonder how the proposed method and DM/Hydra compare when distilled from the parameters obtained from deep ensembles (which are typically far less correlated than the samples obtained from a SGHMC chain).

---

> > > ### Author Response · Authors · 2022-08-10
> > > **Thank you**
> > >
> > > We thank the reviewer for raising the score based on our response. In our experiments we noticed that cSG-MCMC generates correlated samples mainly when sampling from the same mode; however with warm-restarts (as we did in our experiments) we noticed a much higher diversity in the predictions. Indeed, it may be that deep ensemble generates models that are more diverse, but we do not anticipate that the relative order between baselines will change as they are agnostic to the Bayesian method used and only try to imitate its predictions.

---

### Author Response · Authors · 2022-08-01
**General Response**

We thank the reviewers for their helpful and thoughtful feedback. We are encouraged that all reviewers find our work well written. They found it novel (m5GE,a1xo,i7D4), and with significant impact (rtLD). We revised the manuscript according to the remarks made, and will answer here points raised by multiple reviewers:

- Regarding small to mid dataset size: One of the main motivations for the Bayesian approach is that it is less vulnerable to overfitting. As such, most use cases are in the low-data regime which is why we focused on small to medium datasets. Also, if you look at the results of the cSGHMC sampler, the sampled ensemble on Imagenet is as good as running a single model with SGD so there is little motivation to use and distill an ensemble, at least with current MCMC methods.

- Ensemble size: Our main motivation is distilling large ensembles as that is where the inference cost is debilitating. We added a new result in the supplementary (Sec. B.3) that shows that the accuracy plateaus at ~30-40 models. This means that indeed we need more than the 4-8 models commonly used, but perhaps 120 models aren’t optimal.  Additionally, we added an experiment (sec. B.3) on distilling only 8 ensemble models. In this setting, FED outperforms all methods in terms of accuracy as well and archives good results in ECE and NLL; however, all methods suffer from degradation in performance compared to a model distilled from 120 models. It is important to notice that training FED on a large ensemble gives better results, even if only sample a few samples from it in the end.

- Capturing correlations: Being able to return correlations is something approaches such as EnDD cannot do and was one motivation. However, we cannot show the usefulness as it depends on downstream applications using our predictions. As the reviewers said it is presented as a main feature, we corrected the paper only remarking on it once as a motivation.

---

### Meta-Review · Area_Chair_iinp · 2022-08-24

**Recommendation:** Accept
**Confidence:** Certain

**Metareview:**

The paper proposes a method for ensemble distillation, motivated by the need for efficient Bayesian machine learning. The method seems novel and can potentially make a significant contribution to the literature of ensemble distillation and Bayesian machine learning.

The reviewers found the paper well-written and the empirical results compelling. Some concerns were raised about the fairness of the empirical evaluation in the first round of reviews, but these were mostly addressed during the discussion with the authors. Seeing as no major concerns remain, I'm happy to recommend acceptance.


**Award:**

No

---

### Decision · Program_Chairs · 2022-09-14

Accept